# The Use of Portion Control Plates to Promote Healthy Eating and Diet-Related Outcomes: A Scoping Review

**DOI:** 10.3390/nu14040892

**Published:** 2022-02-20

**Authors:** Si Si Jia, Qingzhou Liu, Margaret Allman-Farinelli, Stephanie R. Partridge, Amy Pratten, Lisa Yates, Matthew Stevens, Bronwyn McGill

**Affiliations:** 1Engagement and Co-Design Research Hub, School of Health Sciences, Faculty of Medicine and Health, The University of Sydney, Sydney 2006, Australia; stephanie.partridge@sydney.edu.au; 2Charles Perkins Centre, School of Life and Environmental Sciences, Faculty of Science, The University of Sydney, Sydney 2006, Australia; qingzhou.liu@sydney.edu.au; 3Charles Perkins Centre, Sydney Nursing School, Faculty of Medicine and Health, The University of Sydney, Sydney 2006, Australia; margaret.allman-farinelli@sydney.edu.au; 4Prevention Research Collaboration, Sydney School of Public Health, Faculty of Medicine and Health, The University of Sydney, Sydney 2006, Australia; bronwyn.mcgill@sydney.edu.au; 5New South Wales Ministry of Health, Centre for Population Health, St Leonards, Sydney 2065, Australia; amy.pratten@health.nsw.gov.au (A.P.); lisa.yates@health.nsw.gov.au (L.Y.); matthew.stevens2@health.nsw.gov.au (M.S.)

**Keywords:** portion plate, health promotion, portion size, portion control, healthy eating, population health

## Abstract

The role of portion control plates in achieving healthy diets is unclear. The aim of this scoping review was to systematically map findings from peer reviewed and grey literature to provide evidence for the use of portion control plates to promote healthy eating and nutrition-related knowledge in children and adults. A secondary aim was to review the design characteristics of portion control plates. The search was conducted in four databases, including Medline, CINAHL, Embase, and PsycInfo, and grey literature sources following the PRISMA scoping review guidelines. A total of 22 articles comprising 23 intervention studies and 8 from grey literature were included. It was found that the various two-dimensional and three-dimensional portion control plates examined were effective tools for better portion size selection in healthy children and adults. Most portion control plates dedicated half the plate to vegetables, a quarter to protein, and a quarter to carbohydrates. The use of portion control plates in nutrition interventions appears to promote weight loss among those with overweight and obesity and/or type 2 diabetes. However, portion control plates were mostly used as part of multicomponent interventions and the effectiveness of the portion control plate as a stand-alone educational resource or portion control tool alone was uncertain. Further interventional research is indicated to investigate portion plates as tools to improve dietary behaviours and food consumption at the population level.

## 1. Introduction

Globally, dietary guideline recommendations indicate adherence to a healthy eating pattern, high in healthy food groups and low in energy-dense nutrient-poor foods, is essential for the prevention of mortality and morbidity from diet-related non-communicable diseases (NCDs) [1,2]. A large body of evidence indicates that unhealthy diets low in fruits, vegetables, nuts and seeds, and wholegrains are major contributors to deaths and Disability-Adjusted Life Years (DALYs) [3,4,5,6]. A systematic review with meta-analyses has demonstrated convincing evidence to support healthy dietary patterns with decreased risks of colon and breast cancer, and alternately, an association between unhealthy dietary patterns and increased risk of colon cancer [7]. Furthermore, diets high in refined carbohydrates, sugary drinks, and processed meats have been associated with increased inflammation in the body and can elevate the subsequent risk of heart disease by 46% and stroke by 28% [8]. Moreover, the Mediterranean diet is widely known for its protective effects against coronary heart disease, ischemic stroke, and total cardiovascular disease [9]. This extensively studied dietary pattern promotes consumption of minimally processed plant-based foods such as vegetables, fruits, nuts, legumes, and wholegrain cereals; includes moderate amounts of meat, seafood, and dairy products and olive oil rich in monounsaturated fat; and limits consumption of foods high in saturated fats [9].

Despite clear evidence supporting the consumption of a healthy diet, efforts to combat unhealthy diets in the population have proven to be difficult. Low vegetable intake is common across multiple Western high-income countries such as the USA [10], Australia [11], and the UK [12], and overconsumption of energy-dense nutrient-poor foods is widespread [13]. In Australia, a large cohort nutrition study of almost 200,000 adults showed that people with low-quality diets (lower in healthy plant foods and higher in unhealthy energy-dense nutrient-poor foods) were three times more likely to be obese than those with high-quality diets [14]. Low-cost and convenient foods that are energy-dense and nutrient-poor are prolific in the obesogenic food environment [15], and this may be exacerbated by food industry marketing practices [16]. For the general population, consistent nutritional information that is easy to understand and apply appears to be lacking [17,18].

Dietary guidelines are designed to inform the public of recommended foods and dietary patterns that provide nutrients for optimal health and assist in the prevention of obesity and diet related NCDs [19]. Multiple studies have observed a link between increased knowledge of dietary guidance and increased likelihood of healthy eating patterns in individuals [20,21]. While research suggests that individuals have a relatively high level of basic nutrition knowledge to distinguish healthy and unhealthy foods [22,23], many are confused about more detailed levels of nutrition information. For example, one study demonstrated that only 56 and 62% of adult participants were aware of the recommended number of servings of fruits and vegetables, respectively [24]. Furthermore, nutrition knowledge is often poorly translated into making healthy choices in reality, as studies show that participants lack knowledge about recommended portion sizes [25], and although they are able to evaluate the healthiness of single food items, they have difficulty evaluating entire meals [26]. Dietary recommendations that are applicable to single meal settings are therefore key to helping individuals make healthy food choices. In recent years, dietary guidelines have shifted towards a food-based approach using whole dietary patterns rather than examining nutrients in isolation [27,28]. This ‘food-based’ approach, commonly represented in a visual form [29,30], supports the translation of population nutrition goals into healthy and habitual food choices in individuals [31].

The plate model is a practical nutrition education tool and is now widely incorporated in many dietary guidelines across the globe [32]. It was first proposed by the Swedish Diabetic Association as a simple dietary education method which provides a visual representation of healthy eating through the use of pictures, graphs, charts, and food replicas [33]. The plate is represented as a pie chart which displays appropriate proportions of the plate that should be covered by foods from varying food groups. A major benefit of the plate model is the enhancement of the connection between dietary theory and practice. While two-dimensional (2D) models are widely used to educate the public regarding appropriate portion sizes, three-dimensional (3D) portion plates allow users to apply this dietary theory in practice.

Since 2011, MyPlate has become a primary part of the healthy eating communications initiatives in the United States of America (USA). This 2D plate icon, which represents the five food groups (fruits, vegetables, grains, protein foods, and dairy foods), has been used to communicate the Dietary Guidelines for Americans and teach consumers how to build a healthy plate at mealtimes [34]. Three-dimensional portion control plates that enable people to fill the plate are increasingly used to support weight loss as understanding appropriate sizes of the five food groups are a critical component of weight management [35].

It remains unclear whether portion control plates can improve nutrition knowledge and healthy dietary behaviours. Moreover, to our knowledge, no review has looked at the evidence for portion plate use and healthy eating practices across different population groups including children and adults. The aim of this review was to synthesise findings from peer reviewed and grey literature to provide evidence for the use of portion plates to promote healthy eating and nutrition knowledge in children and adults and to review the design characteristics of portion plates. A secondary aim was to compare portion plate characteristics across varying population groups including healthy children, healthy adults, adults with obesity and/or Type 2 Diabetes (T2DM), pregnant women, and Aboriginal and Torres Strait Islander peoples. Ultimately, the findings from this review may be used to identify gaps to inform the design of useful and culturally appropriate portion plates for varying groups and test their effectiveness in improving population nutrition.

## 2. Materials and Methods

Scoping reviews are a method of evidence synthesis used to map the nature and extent of literature on a topic that has not been comprehensively reviewed and may be used to determine if a systematic review is warranted [36,37]. This scoping review was conducted in accordance with the Preferred Reporting Items for Systematic Reviews and Meta-Analyses for the scoping review process (PRISMA-ScR), as well as the five-stage methodological framework (identifying the research question(s); identifying relevant studies; study selection; charting the data; and collating, summarising, and reporting the results) outlined by Arksey and O’Malley [37]. The review protocol was registered with the Open Science Framework on 8 July 2021 (registration doi: 10.17605/OSF.IO/XK6E7; https://osf.io/xk6e7).

### 2.1. Eligibility Criteria

Eligibility criteria were developed using the participants, concept, context (PCC) framework. Studies must have assessed the use of any type of portion control plate (2D or 3D) to promote healthy eating and nutrition knowledge. Studies that reported acceptability and described characteristics of a portion control plate, as well as those that involved portion control plates as part of multicomponent interventions, were included in this review. Eligible studies were written in English and conducted in high-income Western countries to align with the Australian context. All study populations including adults, children, adolescents, and people with chronic lifestyle-related conditions were included. Quantitative and qualitative peer-reviewed articles as well as grey literature (including commentary, newsletter, conference papers, and book chapters) were included. Studies focused on other portion control tools (for example, portion measuring cups) and did not involve portion plates were excluded.

### 2.2. Search Strategy and Information Sources

A comprehensive search strategy was developed with input from an experienced university librarian. The search was implemented using four electronic databases representing different disciplines (Medline via Ovid, PsycInfo, EMBASE, and CINAHL). The search strategy of Medline via Ovid is included in Appendix A. A citation search was undertaken by scanning the reference lists of eligible studies identified from the database search. Grey literature searches were undertaken using combinations of selected key words within the advanced search functions of Google. These searches were limited to the first 200 results, in keeping with recognised guidance for grey literature searches [38].

### 2.3. Screening and Study Selection

Title and abstract screening, followed by full-text screening were performed using the eligibility criteria by two independent reviewers (SJ, QL). Any discrepancies were discussed between the two reviewers first, and further discussion with a third reviewer (BM) was undertaken if consensus was not reached. The reasons for the exclusion of full text were recorded.

### 2.4. Data Extraction and Presentation of Results

Data extraction was shared by two reviewers (where SJ completed data extraction for half the studies which were checked by QL, and vice versa). Data were extracted using a pre-developed data extraction table. For peer-reviewed intervention studies, summarised data included target population, study design, description of portion control plate, and key findings. Studies reporting changes in nutrition knowledge and dietary behaviours are grouped by target population: 1. adults, 2. children and adolescents, and 3. those reporting the use of portion control plate for weight loss interventions and type 2 diabetes education.

Summarised data of grey literature are presented in a separate table, including target population, branding/manufacturer, size of plate, proportion of major food groups, imagery style, and website link.

### 2.5. Quality Appraisal

All primary studies used to inform the findings were independently assessed for quality by two reviewers (SJ and QL) using the Joanna Briggs Institute (JBI) Critical Appraisal Checklists for the appropriate study type [39]. Any disagreement was resolved by a third reviewer (BM). The risk of bias of individual intervention studies was either rated as high (three or more criteria No or Unclear) or low (less than three criteria No or Unclear) risk of bias [39,40].

## 3. Results

A total of 17 relevant articles were identified through database searching, 5 peer-reviewed articles were identified by performing citation searching, and 8 were identified through the Google searches, resulting in a total of 30 articles (22 peer-reviewed journal articles from selected databases and 8 from the grey literature) included in this review (Figure 1). Most articles were conducted in the United States of America (USA) (*n* = 19), followed by Australia (*n* = 5), the United Kingdom (UK) (*n* = 3), Canada (*n* = 2), and Sweden (*n* = 1). A total of 23 intervention studies were identified from the 22 peer-reviewed articles. Nine were randomised controlled trials (RCTs) (two with low [41,42] and seven with high risk of bias [43,44,45,46,47,48]), seven were quasi-experimental studies (three with low [49,50,51] and four with high risk of bias [52,53,54,55]), five were cross-sectional studies (one with low [56] and four with high risk of bias [57,58,59,60]), and two were qualitative studies with a high risk of bias [61,62].

A narrative summary of the results of intervention studies by population group (adults, children, and people with obesity or type 2 diabetes) is presented in Table 1, Table 2 and Table 3. No studies were specific to Aboriginal and Torres Strait Islander peoples. Of the 23 intervention studies, 16 studies were conducted with adults [41,42,43,44,46,48,49,50,52,53,56,58,59,60,61] and 7 were conducted with children and adolescents [45,47,51,54,55,57,62]. The effect of the portion control plate was examined using 2D plate diagrams (*n* = 12) [44,49,50,52,53,55,56,57,58,59,60,62] and using 3D portion plates (*n* = 11) [41,42,43,45,46,47,48,51,54,61].

An overview of eight types of portion control plates found in the grey literature is provided in Table 4. Five were designed for commercial purposes, while three were developed by non-profit organisations for educational purposes. The portion plates in the grey literature targeted the general population of all age groups (*n* = 2), adults only (*n* = 2), pregnant women (*n* = 2), children aged < 8 years (*n* = 1), and Aboriginal and Torres Strait Islander children aged 2–11 years (*n* = 1).

Based on the JBI Critical Appraisal checklists, 6 studies were rated as low risk of bias [41,42,49,50,51,56] and 17 studies were rated as high risk of bias [43,45,46,47,48,52,53,54,55,57,58,59,60,61,62]. Most of the studies reviewed met the criteria relating to reliable outcome measurements, as well as appropriate intervention design and statistical analysis. Some quantitative studies failed to meet the criteria for reliable exposure and/or outcome measurements. Both qualitative studies demonstrated congruity between the research methodology, the research question, and interpretation of results. However, none met the criteria relating to examination of researchers’ cultural and theoretical orientation during the research process. A summary of risk of bias assessment is attached (Appendix A).

### 3.1. Changes in Nutrition Knowledge and Dietary Behaviours

#### 3.1.1. Adults (*n* = 10)

For a generally healthy adult population, eight of the ten studies examined the effect of portion control plates on their own (Table 1). Four [44,49,59,60] of the ten studies investigated the sole impact of MyPlate in an intervention without accompanying materials; another four studies [46,56,58] exclusively studied the impact of other types of the portion control plate or plate model, such as the Eatwell plate [56], on improving nutrition knowledge and dietary behaviours in adults. The remaining two studies [50,52] included other dietary guidelines such as the Dietary Guidelines for Americans in addition to a portion control plate as part of a nutrition education intervention.

Nine studies reported the efficacy of portion control plates in improving nutrition knowledge and dietary behaviours in adults (aged 18 years and older). Six USA studies used the MyPlate diagram—an educational tool which advocates for the appropriate portion sizes of core food groups [44,49,50,52,59]. Half of MyPlate is designated for fruits and vegetables, a quarter for protein, and a quarter for carbohydrates. MyPlate also includes a serving of dairy as an additional section attached to the plate. Two of these studies using MyPlate [49,50] reported adult participants had increased their nutritional knowledge, with one [50] specifically mentioning its usefulness in educating pregnant women. Two other studies demonstrated that familiarity with MyPlate positively correlated with healthy behaviours such as eating before going grocery shopping [49] and using nutrition information shown on the menu while ordering foods [59]. The majority of studies were completed in a single education session [46,50,56,58] or by using surveys [59,60]. Three studies were conducted over a longer term, with study duration varying from two weeks to two months [44,49,52].

One study examined the use of MyPlate in meal planning and found that the meals of participants who had used MyPlate were lower in total energy, grains, and dairy, yet significantly higher in fruit and vegetables [44]. Likewise, a study used MyPlate as a nutrition education tool as part of its text-messaging intervention and found that the intervention group saw an increase in fruit consumption and a trend towards an increase in vegetable consumption, compared to the control group [52]. Another study from the USA used an alternate portion control plate to MyPlate and observed decreased self-selected portion sizes of protein compared to a regular plate [46]. As part of this study, self-selected portion sizes of vegetables were also shown to be smaller compared to portion sizes plated on a regular plate, and this may counter the expected encouragement of healthy dietary behaviours from use of portion control plates. Another potential negative aspect to portion control plates was highlighted in a Swedish study [58] which found that the plate was perceived as inappropriate for the elderly, difficult to adhere to amounts shown on the plate, and difficult for the majority of individuals to eat large volumes of vegetables.

Despite this, in a cross-sectional study [60], MyPlate was shown to be received favourably by adult participants who found it clear and easy to use. Similarly, the Swedish study revealed that health professionals thought the plate tool was simple, easy to use, useful for many populations, and was easy to understand and memorise [58]. A UK study [56] reported the comparable ‘Eatwell’ plate was also well-regarded by adult participants who rated the plate as a ‘comprehensible graphic display of the Mediterranean dietary guidelines’. Overall, the portion control plate appeared to be effective among adults in increasing nutrition knowledge and supporting positive dietary behaviours such as increased fruit and vegetable intake.

#### 3.1.2. Children and Adolescents (*n* = 7)

Three of seven studies did not involve parallel interventions except the portion control plate [54,57,62], while accompanying interventions were identified in four of seven studies and differed in their design [45,47,51,55]. Regarding intervention design, participants received nutrition-related education prior to the intervention in three studies: one only had a brief orientation [51], and the other two had formal education sessions or standard nutrition counselling on the use of the portion plate [55,57].

Six of seven studies quantitatively assessed the efficacy of portion control plates in improving nutrition knowledge, dietary behaviours, and weight status in children and adolescents by using five different portion control plates [45,47,51,54,55,57]. Of these, two studies were completed in a single education session [45,55], two studies assessing lunch quality were conducted over three days [51,54], one study over a longer period of time (two weeks to examine the change of home food environment) [57], and the remaining study was a weight-loss intervention program conducted over six months [47]. Two studies qualitatively examined the ease of use and preferences of plate characteristics [45,62].

Portion control plates showed positive effects in three out of four studies assessing diet quality [51,54,57]. One study using the MyPlate diagram reported the MyPlate message was associated with higher daily fruit and vegetable intake and diet quality of children aged 8–12 years, as well as better home food environments such as higher home availability of fruits and vegetables [57]. MyPlate was regarded as useful and frequently used by parents from low-income families. The bento-box style Yumbox showed an overall positive influence on foods that caregivers pack for their preschool children. Compared to those who did not use the Yumbox, lunches of children who received Yumbox were found to have a significantly higher variety of components and more fruits, protein, and dairy across three days of observation [51]. One study examined the effect of a segmented plate with fruit and vegetable pictures in young children 3–5 years, demonstrating a significant increase in vegetables selected and consumed over three days, but there were no changes to fruit selected or consumed [54]. The ‘Nutri-plate’ was developed based on adolescents’ feedback [45]. Preferences for portion plate characteristics included recommended portion size and sectors shown on the plate, as well as the use of brighter colours, food icons, and text (educational message) that was easy to understand [45]. In experimental dining sessions offering both healthy and less healthy versions of dishes, for instance, baked or fried chicken for entrée, dining with the Nutri-plate resulted in higher fruit intake and lower overall food intake. Despite this, the Nutri-plate resulted in increased unhealthy broccoli dish consumption and lower steamed broccoli (healthier vegetable choice) consumption [45].

Furthermore, only one study examined the effect of a portion control plate on weight status [47]. Addition of the commercially available ‘Diet plate’ to standard dietary counselling did not result in any significant weight change over six months in children with obesity aged 8–16 years. Regarding nutrition knowledge, one study noted that the MyPlate diagram and message led to a significant increase in nutrition knowledge in low-income middle-school students [55].

#### 3.1.3. Adults with Obesity and/or Type 2 Diabetes

All six interventions which targeted adults with obesity or T2DM incorporated additional accompanying materials to the portion control plate (Table 3) [41,42,43,48,53,61]. Lifestyle advisors [61], dietitians [42], and tele-coaching [41] were used to support participants with the use of the portion control plate. One study also investigated the impact of a guided calibrated crockery set [43] and thus the portion control plate was not exclusively examined. This set included a guided calibrated bowl and glass in addition to the plate. Furthermore, the Idaho plate method (IPM) nutrition program included teaching materials with a teacher’s kit and PowerPoint compact disc in addition to the placemat with the portion plate model illustrated on it [53].

Five quantitative studies and one qualitative study reported the effect of five different portion control plates among adults with obesity or type 2 diabetes (T2DM) [41,42,43,48,53,61]. Two studies were conducted over four weeks [43,61], and four studies were conducted in a longer period of three to six months [41,42,48,53]. All five quantitative studies reported a significant effect of promoting weight loss or blood biomarkers [41,42,43,48]. The guided crockery set (including a guided breakfast bowl, a guided glass, and a guided plate) was found to increase vegetable intake and result in significant weight loss in two weeks [43]. Two studies examining influences of the same glass portion control plate in addition to dietary counselling observed significant weight loss over three months [41,42]. One study [41] found significant reduction in BMI (−0.4 kg/m^2^, *p* = 0.038) and waist to hip ratio (−0.02, *p* = 0.037), and the other study [42] saw a greater percentage of weight change from baseline compared to usual care (−2.4 ± 3.7% vs. −0.5 ± 2.2%; *p* = 0.041) at 3 months. These differences, however, were not significant at six-month follow-up [41,42]. One study noted that, compared to the control group who did not have the Diet plate, those using the Diet plate lost significantly more weight and had better blood lipoprotein profile and glucose control after six months [48]. One study investigated the efficacy of IPM among adults with obesity and T2DM [53], showing that the IPM led to increased intakes of fruit, vegetables, and milk and reduced energy-dense discretionary food consumption over six months. The qualitative study demonstrated strong educational value and high acceptability of calibrated crockery sets [43]. Participants believed that they became aware of the difference between habitual portions and recommended ones and were able to achieve better portion control when using the calibrated crockery set.

### 3.2. Portion Control Plate Characteristics

The design characteristics of portion control plates identified from both peer-reviewed literature and grey literature sources were outlined in this review. Overall, compared to portion control plates found from intervention studies, those found from grey literature reflected similar proportions of major food groups, with a quarter of the plate designated for protein foods, a quarter for carbohydrates, and half a plate of vegetables.

#### 3.2.1. Plate Size, Proportions, Food Groups, and Imagery

Three-dimensional portion plates identified from the grey literature ranged from 9 inches (22.9 cm) to 10 inches (25.4 cm) in diameter (Table 4). On two portion plates, protein was specified to be lean meats (Foodbank and VACCHO) [63,64]. Varying descriptors for carbohydrates were used across portion plates. The Nestlé^®^ Portion plates [65] simply stated, ‘carbohydrates’, while other plates indicated carbohydrates should be ‘Low GI’ (Portion Perfection) [66], ‘wholegrains’ (BeBetter Adult Portion Plate) [67], ‘carbohydrates and wholegrains’ (VACCHO) [63], ‘grain foods’ (Foodbank) [64], ‘wholegrains, legumes, and starchy vegetables’ (My Pregnancy Plate) [68], and ‘grains/starches’ (Healthy Pregnancy Plate) [69].

The Portion Perfection plate specified vegetables to be salad or free vegetables [66]. The VACCHO Healthy Portion Plate and Nestlé^®^ Portion Plate for kids also included salad in the vegetable section [63,65]. Three plates from the USA included fruits in the vegetables section of the plate (BeBetter, My Pregnancy Plate, Healthy Pregnancy Plate) [67,68,69] and this is likely mirroring the USDA MyPlate model.

The ‘Great Ideas in Nutrition Portion Perfection plate’ and ‘BeBetter Adult Portion Plate’ also included a portion for oils and fats in the centre of the plate [66,67]. Furthermore, only the Nestlé^®^ Portion Plate for adults quantified the number of serves on the plate, suggesting consumers place one serve of protein in the corresponding section, two serves of carbohydrates, and three serves of vegetables [65].

Six of the eight portion plates used an ‘artistic’ style for the foods illustrated on the plate [63,64,65,66,67]. The two pregnancy portion plates opted for a more ‘realistic’ representation of foods [68,69]. A white plate background was commonly observed across all portion plates, with the illustrations of foods adding a variety of colours.

#### 3.2.2. Portion Control Plate Characteristics for Children and Adolescents

On the Nestlé^®^ Portion plate for children, different proportions were noted. The plate was divided into a third for vegetables, a third for protein, and a third for carbohydrates [65]. Furthermore, portion plates identified from the peer-reviewed literature which were designed specifically for children and adolescents commonly featured brighter colours, easy to read text, labelled compartments, and heavily emphasised the inclusion of illustrations [45,51,54] (Table 2). The ‘Yumbox’ [51] was a 3D rectangular bento-style box which differed to other 3D portion plates which were typically circular in shape.

#### 3.2.3. Portion Control Plate Characteristics for Weight-Loss Interventions and T2DM

Portion plates used in weight loss interventions and T2DM education were all 3D and were similarly divided into three main sectors (Table 3). However, unlike MyPlate, these portion control plates did not specify the inclusion of fruits in the vegetables section. A difference was also noted for the suggestion of non-starchy vegetables in the designated ‘vegetables’ section of portion plates used in two weight loss interventions [43,61]. One ‘Diet Plate’ contained tape-partitioned sectors for food groups and written affirmations around the rim of the plate [48].

#### 3.2.4. Portion Control Plate Characteristics for Pregnant Women

The ‘My Pregnancy Plate’ from Oregon Health & Science University included detailed information for appropriate food groups and suggested plate proportions (Table 4) [68]. The plate designated a quarter for protein, a quarter for ‘wholegrains, legumes and starchy vegetables’, and the remaining half was split into ‘non-starchy vegetables’ and fruit. Outside of the plate diagram were extra tips for expectant mothers on recommended sources for protein, wholegrains, non-starchy vegetables, and fruits. Additional information on healthy oils, dairy, and physical activity was also provided. The ‘Healthy Pregnancy Plate’ from Kaiser Permanente [69], a Foundation Health Plan of the Northwest in the USA, was similar to the ‘My Pregnancy Plate’ (Table 4). This plate recommended pregnant women use a 9-inch (22.9 cm) plate to guide their portions. The suggested plate proportions included a quarter protein, a quarter grains and starches, and half a plate of fruits and vegetables. Extra information on hydration, mindful eating, and physical activity were also included.

#### 3.2.5. Portion Control Plate Characteristics for Aboriginal and Torres Strait Islander Peoples

One portion plate was designed to be culturally appropriate for Aboriginal and Torres Strait Islander peoples as part of a pilot program to support the health and wellbeing of children of Aboriginal and Torres Strait Islander communities (Table 4) [63]. On this plate, similar portion plate proportions were observed, with half a plate for vegetables, a quarter for lean protein, and a quarter for carbohydrates. In addition, illustrations were Indigenous art commissioned from Aboriginal and Torres Strait Island artists and included imagery of culturally appropriate foods in each section. For example, for ‘lean meat and proteins’, kangaroo and emu meats were displayed.

### 3.3. Primary Uses of Portion Control Plates Identified from Grey Literature

Of the eight portion control plates found from grey literature [63,64,65,66,67,68], five were commercially available and three were designed for educational purposes from non-profit organisations (Table 4).

#### 3.3.1. Portion Control Plate for Commercial Purposes

There were five portion plates commercially available including the ‘Portion Perfection’ plate from Great Ideas in Nutrition, the ‘Adult Portion Plate’ from BeBetter, and the two ‘Nestlé^®^ Portion Plates’ for children and adults [64,65,66]. These ranged in price from 3.95 AUD for the Nestlé^®^ Portion Plates [65] to 29 AUD for the BeBetter ‘Adult Portion Plate’ [67]. Foodbank WA, a not-for-profit organisation, sold its portion plate for 10 AUD each [64]. Most plates were made of melamine with the ‘Portion Perfection’ plate having a ‘porcelain’ option. Of the five commercial plates, two were specifically targeting weight management (Portion Perfection and Adult Portion Plate) [66,67] and the others (Foodbank and Nestlé^®^ Portion Plates) [64,65] were mostly a resource for general nutrition education. The ‘Portion Perfection’ plate was also marketed as designed by a dietitian with over 25 years of experience in weight management [66]. In a 2018 Nestlé^®^ Australia submission to the Select Committee into the Obesity Epidemic in Australia, Nestlé^®^ Portion Plates were part of Nestlé^®^’s work in ‘boosting understanding about nutrition’. The company described these portion plates as practical resources and educational tools for both healthcare professionals and members of the community [70].

#### 3.3.2. Portion Control Plate Developed by Non-Profit Organisation

The three portion plates created from non-profit organisations [63,68,69] were mainly 2D diagrams designed to educate target populations on appropriate portion sizes of food groups. As these were mostly diagrams, information on plate sizes were not found. These plates were distributed as a nutrition education resource and often used as part of health promotion programs.

## 4. Discussion

This review adds to the body of evidence on the impact of portion control plates on promoting healthy eating and nutrition knowledge, in addition to weight loss outcomes. The portion control plate has been used as both a 2D educational tool to guide dietary behaviours and as a 3D plate used at mealtimes to guide consumption. Overall, portion control plates appeared to increase nutrition knowledge and supported positive dietary behaviours such as increased fruit and vegetable intake in children and adults. In addition, these tools appeared to support weight loss in individuals with obesity and/or T2DM and were highly acceptable and easy to use. Despite these promising results, in most studies, portion control plates formed part of a multi-component intervention and were not examined in isolation. Insufficient evidence in the published peer-review literature was found on the use of portion plates to promote health in key priority population groups such as pregnant women and for Australian Aboriginal and Torres Strait Islander peoples.

Findings from this review support the efficacy of portion control plates as both a conceptual 2D model and as a 3D plate used during mealtimes. Across the included studies, varying versions of the portion control plate commonly accommodated half the plate for vegetables, a quarter for lean protein, and a quarter for carbohydrates. These proportions align with the original plate model promoted by the Swedish Diabetic Association since 1987 [71]. As this review has shown, the 2D model is effective for improving nutrition knowledge in both children and adults. It is suggested that the plate model allows individuals to tailor their meals more flexibly according to the marked proportions [33]. Moreover, individuals may respond better to the portion plate’s use as a motivational tool to guide meal planning. The plate’s flexibility for food choices may be more encouraging than a strict diet based on exact amounts of nutrients, energy, or servings of food groups to achieve each day. In addition, this review suggests that the portion control plate as a 2D model was found to be primarily used in dietary guidelines for Americans and for the UK. A similar plate model is also used in Singapore’s ‘MyHealthyPlate’ [72]. This demonstrates the capacity for portion control plates as a visual tool that can communicate healthy eating habits in an easy-to-understand manner.

Portion control plates also proved to be effective in supporting weight loss for individuals with overweight or obesity. These results align with a recent systematic review and meta-analysis which showed that portion control plates significantly reduced body weight by 2.02 kg (95% CI −3.03 to −1.01, *p* < 0.0001) and body mass index by 0.87 kg/m² (95% CI −1.28 to −0.47, *p* < 0.0001) [73]. As a 3D model, the portion control plate possibly works by reducing the amount of food consumed, thereby resulting in weight loss through caloric deficit. A Cochrane review conducted by Hollands and colleagues confirmed a small to moderate effect of larger tableware on increased food intake [74]. Furthermore, portion plates may be a source of external cues for individuals affecting their meal planning, food selection [75], and hence food intake.

For a generally healthy adult population, our findings suggest that portion control plates may be effective on their own and do not need accompanying materials to show improvements in nutrition knowledge and dietary behaviours. In contrast, portion control plates targeted at adults with obesity or T2DM often were all supported by a dietitian or lifestyle advisor. Addressing overweight and obesity is complex and weight loss from lifestyle interventions is difficult to achieve and sustain [76]. As such, interventions with a wide array of accompanying materials may be essential for significant improvements to health outcomes. A systematic review conducted by Maula and colleagues [77] has shown efficacy of interventions that incorporate education together with a weight loss diet in overweight and obese adults with T2DM. Education combined with low-calorie, low-carbohydrate, or low-fat meal replacements or diets was found to achieve the largest reduction in weight and BMI [77]. For children and adolescents, portion control plates on their own appear to be an effective tool to promote better diet quality and increased vegetable intake. Unlike the interventions for adults, parallel interventions for children and their caregivers tended to be brief and were primarily around providing instructions on the use of portion plates [45,51,54,55]. The success of such interventions might be due to the portion plate itself disseminating food-focused educational messages and also acting as an environmental cue to facilitate appropriate portion size selection [78]. However, interventions that involved the portion plate and standard nutrition counselling did not show any positive effect on weight loss in children [47]. It might be that the portion control plate is one of many ‘narrow’ interventions targeting portion control. There are numerous environmental factors (for example, screen watching while eating) and lifestyle factors that influence weight management among children [79]. Broader lifestyle approaches may be more effective to decrease childhood obesity [79,80].

There is a paucity of published peer-reviewed literature on portion control plates that were tailored to key priority groups for nutrition interventions such as pregnant women and Aboriginal and Torres Strait Islander peoples. An evidence review has similarly shown that few nutrition education programs in Aboriginal and Torres Strait Islander communities have been evaluated [81]. This is concerning when, globally, the health disparities between Indigenous and non-Indigenous populations are pervasive [82]. Furthermore, this review did not find any studies conducted in high-income Western countries on portion plates that were tailored to culturally and linguistically diverse groups. In a Singapore quasi-experimental study conducted by de Korne and colleagues [83], a varied portion plate design was used to accommodate various cultural tastes. This involved a ‘mix’ portion between the lines demarcating vegetables and protein as many Singaporean meals and other Asian foods have these two components pre-mixed [83]. In countries with vibrant migrant communities and high degrees of ethnic diversity, the inclusion of such adaptable plate design features warrants further consideration.

We found comparable results to other previously published reviews. For example, a review conducted by Vargaz-Alvarez and colleagues found portion control tools marginally induced weight loss [84]. However, these tools were inclusive of bowls and cutlery in addition to portion control plates. The findings also concluded that these portion control tools may only have a positive size manipulation of foods when they are used alongside other tools. Likewise, Almiron-Roig and colleagues examined effective strategies to reduce portion sizes [85]. They found that portion size intakes were more consistent when calibrated (partitioned or portion-controlled) plates were used when compared to other types of modified tableware such as the size and shape of plates, bowls, cutlery, or glasses.

### Implications, Strengths, and Limitations

Researchers, clinicians, and government organisations can consider the use of portion control plates as a healthy eating tool to improve population diets. Findings from this review can be used to improve the design of current portion control plates and develop new plates that are effective for different population groups. This review adds to the literature by synthesising the evidence on the health promotion potential of portion control plates to increase nutrition knowledge and other health-promoting behaviours. Furthermore, we conducted a comprehensive grey-literature search on other available portion control plates not reported in the published peer-reviewed literature.

Despite this, we acknowledge limitations with this scoping review. Firstly, the majority of papers (17 out of 23 studies) included in this scoping review were classified as having a high risk of bias. As such, caution is warranted when interpreting the findings from our review as the small body of evidence available is limited by the lower quality of studies. This also suggests that future research on portion plate interventions for health promotion should strive to minimise risk of bias by strengthening study design at all stages. Furthermore, due to the nature of this scoping review, only studies that were conducted in Western, high-income countries were included, and most of the included studies were from the USA. Relevant studies from non-Western countries, for example, Asian and Middle Eastern countries, may bring valuable insights into the development of culturally appropriate portion control plates for use in Australia’s multicultural setting. In addition, only studies written in English were included. This could have excluded some relevant studies written in other languages. However, due to the limited research available on portion plates and health promotion, we believe that this exclusion had minimal impact. Website searching was comprehensive but was limited to the first 200 results, thereby narrowing the potential results found. This is, however, in accordance with the suggested strategy for grey literature searches [38]. Moreover, the efficacy of portion control plates was not examined in isolation as portion plates were commonly involved as one part of a nutrition intervention or health education in included studies. The usefulness of portion control plates alone warrants further investigation. Future intervention studies looking at portion control plates should include control groups who receive portion control plates without any Appendix A such as nutrition education or support.

## 5. Conclusions

Portion plates appear to be a promising tool with studies showing improvements to healthy dietary behaviours and an increase in nutrition knowledge in both children and adults. Most portion control plates followed a common proportion dedicating half the plate to vegetables, a quarter to protein, and a quarter to carbohydrates. For children and adolescents, more imagery and artistic illustrations were used. For individuals with obesity and/or T2DM, portion control plates appeared to be effective when they were used in addition to supporting nutrition interventions. Moreover, most studies did not directly assess the effectiveness of portion control plates in isolation as a tool that could improve adherence to dietary guidance. Many involved the portion plate model as one component of a study or intervention. In addition, most of the research found supports the effectiveness of the portion control plate in changing portion size selections—the intended intake—and few studies reported on the consumption of foods placed on the plate—the actual intake. Thus, the evidence available is insufficient to confirm the usefulness of portion control plates with a high level of certainty. Caution is therefore warranted in interpreting the findings from this review. Furthermore, the usefulness and efficacy of these portion control plates needs to be investigated for some priority groups.

## Figures and Tables

**Figure 1 nutrients-14-00892-f001:**
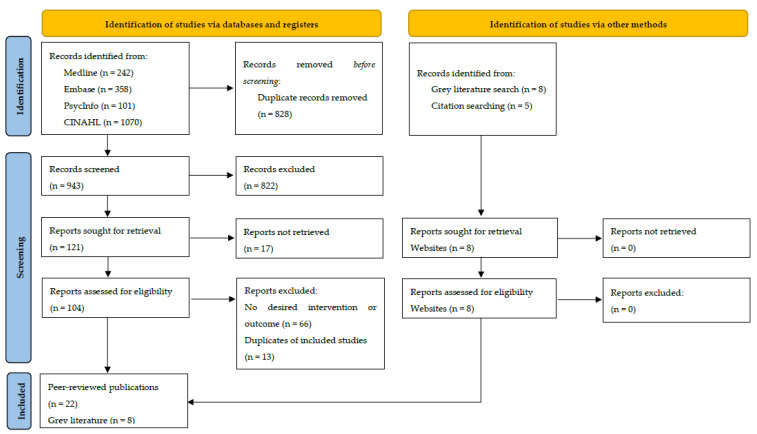
Preferred Reporting Items for Systematic Reviews and Meta-Analyses for the scoping review process (PRISMAScR) flow diagram.

**Table 1 nutrients-14-00892-t001:** Studies reporting changes in nutrition knowledge and dietary behaviours in adults.

First Author, Year, Country	Target Population	Study Design, Risk of Bias	Description of Portion Control Plate	Key Findings
Amaro 2017 [49] USA	Latinas (grocery shopping)	Quasi-experimental, Low	2D MyPlate diagram: 1/2 plate FV, 1/4 plate of grains, 1/4 plate of protein, and 1 serve of dairy.	↑ Knowledge, self-efficacy, and behaviour related to food purchasing. These were maintained for 2 months in a convenience sample of Latinas who viewed videos based on MyPlate.
Bachman 2016 [44] USA	Adults 18–65 y	RCT-Crossover, High	2D MyPlate diagram: 1/2 plate FV, 1/4 plate of grains, 1/4 plate of protein, and 1 serve of dairy.	Meals planned using MyPlate ↓ in total energy, grains, and dairy, and significantly ↑ in FV compared to the 2010 Dietary guidelines.
Blondin 2018 [50] USA	Pregnant women	Quasi-experimental, Low	2D MyPlate diagram: 1/2 plate of FV, 1/4 plate of grains, 1/4 plate of protein, and 1 serve of dairy.	↑ Nutrition knowledge by 17% in pregnant women following the nutrition education session (including the MyPlate guidelines).
Brown 2014 [52] USA	University students	Quasi-experimental, High	2D MyPlate diagram: 1/2 plate FV, 1/4 plate of grains, 1/4 plate of protein, and 1 serve of dairy.	The intervention group saw ↑ in fruit consumption and showed trend towards ↑ vegetable consumption, compared with the control group.
Hughes 2017 [46] USA	General population	Study 1: RCT-Crossover, High	3D Portion control plate: a 25 cm foam plastic plate with a border of 2 cm, had serving size indicators for FV, grains/starches, and protein.	The self-selected portion size of protein was ↓ on the portion control plate compared to the regular plate. No significant difference was observed for grains and vegetables.
	General population	Study 2: RCT-Crossover, two studies assessing lunch quality were conducted over three days, High	3D Portion control plate: a 25 cm foam plastic plate with a border of 2 cm, had serving size indicators for FV, grains/starches, and protein.	The self-selected portion sizes of protein, vegetables, and grains were ↓ on the portion control plate compared to the regular plate.
Lara 2015 [56] UK	Older adults age >50 y	Cross-sectional, Low	2D Eatwell plate diagram: 5 portions of FV, 1/3 daily food starchy foods, some dairy, some protein sources, and small amounts of foods and drinks high in fat and sugar.	British older adults rated the plate as comprehensible graphic displays of Mediterranean diet guidelines.
Nydahl 1993 [58] Sweden	Health professionals	Cross-sectional, High	2D Plate model: plate divided into 3 sections; smallest part (1/4 plate) represents proteins, other two parts are equal size, representing FV and starch.	Plate tool favourably received by majority of participants. Perceived advantages: simple, concrete tool, easy to use, useful for many populations, message easy to understand and memorise. Disadvantages: not appropriate for elderly, difficult to (a) follow amounts shown on plate, (b) to eat such large amounts of vegetables.
Tagtow 2017 [59] USA	General population aged over 16	Cross-sectional, High	2D MyPlate diagram: 1/2 plate of FV, 1/4 plate of grains, 1/4 plate of protein, and 1 serve of dairy.	Familiarity with MyPlate appears to be positively correlated with self-perceived diet quality. Familiarity with MyPlate is positively associated with using nutrition information on the menu while ordering foods.
Wansink 2013 [60] USA	Adult women (≥2 children at home)	Cross-sectional, High	2D MyPlate diagram: 1/2 plate of FV, 1/4 plate of grains, 1/4 plate of protein, and 1 serve of dairy.	Early MyPlate adopters found it clear and easy to use, MyPlate familiarity was highest among those who found it easy to understand and who were also familiar with MyPyramid.

FV = fruits and vegetables.

**Table 2 nutrients-14-00892-t002:** Studies reporting changes in nutrition knowledge and dietary behaviours in children and adolescents.

First Author, Year, Country	Target Population	Study Design, Risk of Bias	Description of Portion Control Plate	Key Findings
Shukaitis 2021 [51] USA	Preschool children	Quasi- experimental, Low	3D Yumbox: bento-style box with labelled compartments (vegetable, fruit, dairy, grain, protein).	Yumbox had a positive influence on the types of foods caretakers pack for their children’s lunches (↑ variety of components, ↑ fruit, ↑ protein, and ↑ dairy across all 3 days, and ↑ vegetables for the first 2 days).
Arcan 2019 [57] USA	Children 8–12 y and caregivers	Cross-sectional, High	2D MyPlate diagram: 1/2 plate of FV, 1/4 plate of grains, 1/4 plate of protein, and 1 serve of dairy.	The ’half plate FV’ message was positively associated with daily FV intake, diet quality, and home food environment (additional weekly family meal, higher parent and child cooking skills, higher home availability of FV).
Melnick 2018 [54] USA	Preschool children 3–5 y	Quasi-experimental, High	3D Segmented plate: plates with segments of FV sections, FV pictures in designated sections.	Segmented plates with FV pictures resulted in a significant ↑ vegetables taken and vegetables consumed, no significant changes to fruit taken or consumed) over 3 days.
Ho 2016 [47] Canada	Children 8–16 y, BMI > 85 percentile	RCT, High	3D Diet plate: commercially available dinner plate with tape-partitioned sections for carbohydrates, proteins, sauces; remainder of plate for vegetables.	Addition of a portion control tool to standard nutritional counselling did not result in a significant change in BMI z score, anthropometric measures, or laboratory markers after 6 months compared to standard nutrition counselling.
Shilts 2015 [62] USA	Low literate, low-income families	Qualitative, High	2D MyPlate diagram: 1/2 FV, 1/4 grains, 1/4 protein, and 1 serve of dairy.	The My Healthy Plate lessons improved parents’ child feeding-related behaviours. My Healthy Plate was perceived as useful and frequently used by parents. A fast-food poster showing common fast-food options (pizza, hamburger, and taco) in MyPlate proportions was developed based on parents’ feedback.
Ellsworth 2014 [55] USA	Low-income middle-school students	Quasi-experimental, High	2D MyPlate diagram: 1/2 plate of FV, 1/4 plate of grains, 1/4 plate of protein, and 1 serve of dairy.	↑ Nutrition knowledge in middle school students who used a mobile farmers’ market and received nutrition education that involved MyPlate guidelines.
Bohnert 2011 [45] USA	African American adolescents	RCT, High	3D Nutri-plate: brighter colour balanced with more neutral ones; text that was easy to read; designated sections for vegetables, protein, wholegrains, oils; designated visual representations in sections for vegetables, protein, wholegrains, oils.	Quantitative findings: Dining with the Nutri-plate did not appear to influence healthy food selected by participants overall. The Nutri-plate appeared to facilitate ↑ fruit, ↑ unhealthy broccoli, ↓ steamed broccoli, ↓ overall food. Qualitative findings: Participants’ preferences for plate characteristics including: (i) space to put food shown; (ii) recommended portion size shown; (iii) proportion of food groups shown; (iv) brighter colours, food icons, text that was easy to read, and written messages about healthy eating.

FV = fruits and vegetables.

**Table 3 nutrients-14-00892-t003:** Studies reporting the use of portion control plate for weight loss interventions and type 2 diabetes education.

First Author, Year, Country	Target Population	Study Design, Risk of Bias	Description of Portion Control Plate	Key Findings
Almiron-Roig, 2016 [43] UK (Phase 2)	Adults with obesity	RCT-Crossover, High	Guided calibrated crockery set: guided bowl, guided glass, and 3D 23 cm guided plate (three sectors: 1/2 non-starchy vegetables, 1/4 protein, 1/4 plate starch).	Both portion control tools led to significant weight loss and positive dietary behaviours (↑ raw and cooked vegetables, ↓ potatoes and chips) among obese population. Tool type did not show a significant difference on outcome measures.
Almiron-Roig, 2019 [61] UK	Adults with obesity	Qualitative, High	Guided calibrated crockery set: guided bowl, guided glass, and 3D 23 cm guided plate (three sectors: 1/2 non-starchy vegetables, 1/4 protein, 1/4 plate starch).	Strong educational benefits were clearly identified: learned appropriate portion sizes, became aware of difference between habitual portions and recommended ones, useful visual aid to compare portions against.
Edens, 2013 [53] USA	Adults with T2DM	Quasi-experimental, High	3D Idaho plate method (IPM): a meal-planning education program for T2DM consumers, including a colourful-illustrated foam placemat (1/2 plate vegetables, 1/4 plate bread/starch/grain, 1/4 plate meat/protein).	The IPM was frequently used in meal planning for all meals, leading to ↑ FV and milk, ↓ high-fat energy dense foods (fried potatoes, French fries, margarine or butter, hotdog.
Huber, 2015 [41] USA	Adults with obesity	RCT, Low	3D Portion control plate: a clear glass plate with black print dividing it into three sections (1/2 ‘vegetables’, 1/4 ‘fish, lean meat, chicken, nuts’, 1/4 ‘potatoes, pasta, rice, beans and wholegrains’).	Participants in intervention group (tele-coaching and portion control plate) had a significant weight loss at 3 months (compared to baseline), but this difference was not significant at 6 months.
Kesman, 2011 [42] USA	Adults with obesity	RCT, Low	3D Portion control plate: clear glass plate with black print dividing it into three sections (1/2 ‘vegetables’, 1/4 ‘fish, lean meat, chicken, nuts’, 1/4 ‘potatoes, pasta, rice, beans and wholegrains’).	Participants in the intervention group (dietary counselling and portion control plate) had a more significant weight reduction at 3 months compared to those in the control group; this difference was not significant at 6 months.
Kline 2007 [48] Canada	People with obesity	RCT, High	3D Diet plate: commercially available dinner plate with tape-partitioned sections for carbohydrates, proteins, and sauces, remainder of the plate for vegetables, plus a breakfast bowl.	Participants in the intervention group lost significantly more weight and had a significantly greater decrease in non-HDL lipoprotein than control group. A greater proportion of intervention group participants required a decrease in hypoglycaemic medication/had a significant decrease in daily insulin dose use at 6 months.

FV = fruits and vegetables.

**Table 4 nutrients-14-00892-t004:** Grey literature reporting characteristics of portion control plates.

Name of Portion Control Plate/Country	Target Population	Branding/Manufacturer	Proportions/Serves/Portions of Major Food Groups	Style of Imagery/Key Messages/Website Link
Portion Perfection. Australia.	General population.	Designed by a dietitian. Great Ideas in Nutrition. Commercial.	1/4 protein, 1/4 low GI carbs, 1/2 salad or free veg, 1 tsp oil or 1 tbsp low fat sauce.	Art imagery: arrows delineate portions. Rim: strategic reminders on plate border for meal satisfaction with Enjoy! Presentation, Variety, Aromas, Textures, Temperature, Flavours. https://www.greatideas.net.au/portion-perfection-plate-melamine.html, accessed 15 July 2021
The Adult Portion Plate. United States of America.	Adults.	BeBetter: Take a better look at your portions. Commercial.	1/4 meat/proteins, 1/4 wholegrains, 1/2 FV, 1 central portion for fats/oils.	Realistic imagery. Rim: icons engaging in physical activity such as weightlifting, golf, stretching, hula-hooping. https://www.bebetter.com/press_2011.06.02.html, accessed 15 July 2021
Growing up healthy and deadly ‘Healthy Portion Plate’. Australia.	Aboriginal and Torres Strait Islander Children.	Victorian Aboriginal Community Controlled Health Organisation. Non-profit.	1/4 lean meat/proteins, 1/4 carbs/wholegrains. 1/2 veg/salads.	Indigenous art imagery. https://www.vaccho.org.au/wd/nutrition/guhd/, accessed 15 July 2021
Nestlé^®^ Portion Plate for Kids. Australia.	Children < 8 years.	Australian Institute of Sport and Nestlé^®^ Healthy Active Kids. Commercial.	1/3 veg/salads, 1/3 protein, 1/3 carbs.	Art imagery. Rim: What’s on your plate kids? Healthy Eating is as easy as 1,2,3. Salad and Veggies section: Keep it Colourful https://shop.ais.gov.au/Nestle-Portion-Plate, accessed 15 July 2021
Nestlé^®^ Portion Plate for Adults. Australia.	Adults.	Nestlé^®^ Choose Wellness: Good Food, Good Life. Commercial.	1/4 protein (1 serve), 1/4 carbs (2 serves), 1/2 veg (3 serves).	Art imagery. Rim: Know your Portions. Protein section: Choose lean cuts of meat, trim visible fat and take skin off chicken. Carbohydrate section: Choose Low GI or wholegrain varieties. Vegetables section: Choose variety of colours. https://shop.ais.gov.au/Nestle-Portion-Plate, accessed 15 July 2021
Foodbank WA Portion Plate. Australia.	Low-middle income general population.	Foodbank WA. Non-profit.	1/4 lean meat and alternatives, 1/4 grain foods, 1/2 veg.	Art imagery. https://www.superherofoodshq.org.au/foodsensations/product/portion-plate/, accessed 15 July 2021
My Pregnancy Plate United States of America	Pregnant women	Oregon Health & Science University. Non-profit.	1/4 protein, 1/4 wholegrains, legumes, 3/8 non-starchy veg, 1/8 fruit.	Realistic imagery. Recommendations for types of proteins, wholegrains, starchy and non-starchy veg and fruit to choose. Recommendations for healthy oils and dairy. https://www.ohsu.edu/womens-health/my-pregnancy-plate, accessed 15 July 2021
Healthy Pregnancy Plate United States of America	Pregnant women	Kaiser Permanente, Foundation. Non-profit.	1/4 protein, 1/4 grains/starches, 1/2 FV.	Realistic imagery. Recommendations for practicing mindful eating, hydration, physical activity, and limiting added sugar. https://healthy.kaiserpermanente.org/content/dam/kporg/final/documents/health-education-materials/instructions/healthy-pregnancy-plate-hi-en.pdf, accessed 15 July 2021

## Data Availability

Not applicable.

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
