# Peer review of "The Use of Portion Control Plates to Promote Healthy Eating and Diet-Related Outcomes: A Scoping Review"

_nutrients, 2022, doi:10.3390/nu14040892_

Round 1

Reviewer 1 Report

The authors carried out a scoping review on a relevant topic for public health and community nutrition. The paper is well structured and written; its language is clear and concise. The supplementary material complements the information required to review the paper. The methodology is appropriately described including the eligibility criteria, the search strategy and the information sources. Two of the authors independently assessed the quality of each paper with critical appraisal checklists according to study type. Each paper included in the review was therefore rated for the risk of bias: 17 out of 23 interventions (include in 22 papers) were rated "high risk" and only 6 had a low risk of bias. The authors identify the strengths and weaknesses of each paper. In general, the results are appropriately discussed and the conclusion follows logically. However the quality appraisal is not fully addressed as the authors did not take into account the fact that the majority of papers were classified of "high risk of bias". This is to be included in the discussion in order to improve the manuscript. 

In addition to this, references 68 and 69 seem to be uncomplete; ref 71 was not available from the link provided.

Author Response

Thank you for your comments. We have added in the high risk of bias into the discussion as a major limitation to our findings. Please see lines 1443-1448 in the revised manuscript in section 4.1 of the Discussion.

We have also fixed references 68, 69 and 71 – now references 69, 70 and 72.

Reviewer 2 Report

This is an interesting review of the subject, which provides a very useful analysis when choosing an easy-to-apply nutrition education tool. The most important problem is stated by the authors, since they only include studies performed in Western countries of good socioeconomic level and exclude possible studies performed in important populations both in size and in problems related to poor nutrition, such as Asia, Eastern Europe and Latin America. Including some studies from these places (despite the language) would have greatly enriched the results and the discussion. Some comments: Line 173, figure 1 is mentioned, however it is not found in the main text or in the supplementary material to which I had access. Line 252, it is misconfigured, it should be placed as a subtitle Line 269, references are as superscript Line 296, it is misconfigured, it should be placed as a subtitle Line 314, references are as superscript Line 364, references are as superscript

Author Response

Thank you for your comments. We accidentally omitted Figure 1 in our first submission – please see the PRISMA diagram on page 6 of the revised manuscript. We have also corrected all formatting issues.